# A psychometric evaluation of the Chinese Impact of Vision Impairment (C-IVI) questionnaire in an adult cohort with high myopia using Rasch analysis

Wei Pan[1], Ryan E. K. Man[2,3], Eva K. Fenwick[2,3], Ziqi Hu[4], Yanfeng Jiang[1], Huanhuan Tan,[4], Zhongping Chen[4], Quan V. Hoang[2,3], Seang-Mei Saw[,2,3,5], Ecosse L. Lamoureux[2,6], Weizhong Lan[1,4,7,8]*, Zhikuan Yang[1,4,8]

1 Aier Academy of Ophthalmology, Central South University, Changsha, China, 2 Singapore Eye Research Institute, Singapore National Eye Centre, Singapore, 3 Duke-NUS Medical School, Singapore, 4 Changsha Aier Eye Hospital, Changsha, China, 5 Saw Swee Hock School of Public Health, National University of Singapore, Singapore, 6 University of Melbourne, Australia, 7 Hefei Aier Eye Hospital, Anhui Medical University, Hefei, China, 8 Aier School of Optometry and Vision Science, Hubei University of Science and Technology, Xianning, Hubei, China

* lanweizhong@aierchina.com

## Abstract

### Purpose

A psychometric evaluation of the Chinese Impact of Vision Impairment (C-IVI) questionnaire in an adult cohort with high myopia using Rasch Analysis and determination of the relationship between vision-related quality-of-life (VRQoL) and myopia macular degeneration (MMD).

### Methods

We used the baseline visit data of the AIER-Singapore Eye Research Institute (SERI) High Myopia Adult Cohort Study. VRQoL was assessed using the 28-item C-IVI. Rasch analysis was conducted to evaluate the overall C-IVI and domain scores ('Mobility and independence'—MB, 'Reading and accessing information'—RD, and 'Emotional well-being'—EWB), including response category functioning, precision, unidimensionality, targeting, and differential item functioning (DIF). The criterion validity, C-IVI's ability to distinguish participants based on severity of vision impairment (VI), spherical equivalent (SER), and the presence of MMD were analyzed using ANOVA and pairwise t-tests.

### Results

There were 431 participants, with mean (SD) age of 42.2 (7.1) years, SER of −8.3 (3.8) D, and visual acuity of 0.1 (0.2) LogMAR. Of these, 15.8% presented MMD, 79.4%, 13.5%, 7.0%, and 0.2% had no, mild, moderate, and severe VI, respectively.

**Data availability statement:** Deidentified data supporting the findings of this study is shared via Kaggle, a public available data repositories platform. Consent for publication of raw data obtained from study participants. Data is shared under CC0 License. Data Link please see: https://www.kaggle.com/datasets/panwei052117/ivi-questionnaire-responses-aier-seri.

**Funding:** The authors declare no conflicts of interest. This study was funded by the Science and Technology Innovation Program of Hunan Province, China (2024RC5002, 2023RC1079). The funders had no role in research design or decisions in publications. The authors alone are responsible for the content and writing of the paper.

**Competing interests:** The authors have declared that no competing interests exist.

Response thresholds were ordered for the overall and three domains. The overall range-based precision was 0.94, and 0.80 for each domain. The three domains demonstrated unidimensionality. DIF was uniform in overall and EWB, but not the MB and RD domains. Person estimates decreased with increasing VI severity, worsening SER, and presenting MMD (all p < 0.05) for the overall and domain scores.

## Conclusions

The C-IVI questionnaire is a valid and reliable tool for assessing VRQoL in adults with high myopia in China.

## Introduction

Blindness is consistently ranked among the most feared health conditions globally [1,2]. Myopia, particularly high myopia, significantly increases the risk of developing serious ocular conditions, including myopic macular degeneration (MMD), that can lead to severe visual impairment (VI) or blindness [3]. As myopia progresses, uncorrected visual acuity (VA) deteriorates markedly. For example, an individual with a spherical equivalent (SER) of −5.00D typically has a uncorrected VA of around 6/172 (Snellen), equivalent to 1.46 LogMAR—well below the threshold for blindness (3/60 Snellen, or 1.30 LogMAR) [4].

Alarmingly, it has been predicted that by 2050, 50% of the global population will be myopic (SER ≤ −0.50D), with 10% classified as highly myopic (SER ≤ −5.0D) [5]. The myopia epidemic is particularly pronounced in East Asia. In China, the current prevalence of childhood myopia is 36.6%, with projections suggesting it will rise to 61.3% by 2050 [6]. Moreover, by the same year, the prevalence of high myopia among children is expected to reach 18.8% in urban areas [6]. The Beijing Eye Study, a population-based study, found that the prevalence of moderate/severe visual impairment (MSVI) was 18.9%, the prevalence of blindness was 4.7% among 212 high myopia adults, and the leading cause of MSVI and blindness was MMD (accounting for 58% of the cases) [7]. If these projections hold, a significant proportion of the Chinese population may experience moderate to severe VI or blindness in adulthood.

The impact of vision impairment (IVI) questionnaire is a vision-related quality-of-life (VRQoL) questionnaire that assesses the impact of visual impairment (VI) from the patients' perspectives [8–11]. Originally developed as a 32-item instrument, a shorter 28-item version has become more widely used [8–11]. This questionnaire has been translated into various languages [11–15], including Chinese (C-IVI) [11]. The translation and validation of the C-IVI were carried out as part of the Singapore Chinese Eye Study, led by the Singapore Eye Research Institute [11]. The 28-item IVI has demonstrated robust psychometric properties and has been extensively validated in populations with different degrees of VI and a range of severe ocular conditions, such as age-related macular degeneration, glaucoma, cataracts, and diabetic retinopathy [16–19]. However, its application in adults with high myopia, and MMD has not yet been explored.

Therefore, this study aims to evaluate the psychometric properties of the of the C-IVI overall and 3 domain scores in a cohort of adults with high myopia in China, using Rasch analysis. Additionally, the study seeks to compare the VRQoL between adults with high myopia associated with MMD and those without MMD.

## Methods

Participants were drawn from the AIER-Singapore Eye Research Institute (SERI) High Myopia Adult Cohort Study, an on-going community-based cohort study conducted in Changsha, China. The study's design and baseline data were previously published [20]. Briefly, the cohort study aims to understand the natural progression, complications of high myopia, and risk factors to complications (e.g., MMD). The current study is a cross-sectional analysis of baseline data from the AIER-SERI High Myopia Adult Cohort Study. The study obtained ethics approval from the Centralized Institutional Review Boards of the Aier Eye Hospital, Changsha and was conducted in accordance with the tenets of the Declaration of Helsinki. Written informed consent was obtained from all subjects.

### Participants

Participants were recruited between August 2021 and December 2023. A total of 445 participants were enrolled at the Changsha Aier Eye Hospital in Hunan Province, China, through social media outreach and word-of-mouth referrals within the community. Inclusion criteria were: 1) adults aged over 30 years, and 2) presence of high myopia (SER ≤ −5.00D). Fourteen participants were excluded due to: 1) severe ocular pathology (strabismus and amblyopia, N = 1; glaucoma, N = 3), 2) incomplete visual acuity testing (N = 2), and 3) incomplete C-IVI questionnaire responses (N = 8).

### Sociodemographic and systemic health data

Sociodemographic and systemic health data were collected using self-administered questionnaires. The sociodemographic variables included in the analysis were age, gender, ethnicity, and educational qualifications. Systemic diseases assessed included hypertension and diabetes.

### Ocular assessment

Each participant underwent a comprehensive ocular examination, including measurements of refractive error, axial length, and visual acuity. Refractive error was measured by an autorefractor (Nidek ARK-1, Nidek Technologies, Gamagori, Japan) after cycloplegia. Cycloplegia was achieved with 2 drops of 1% cyclopentolate (Cyclogyl, Alcon, Geneva, Switzerland) following 1 drop of Alcaine (Proparacaine Hydrochloride 0.5%, Alcon, Geneva, Switzerland). Refractive error was calculated as spherical equivalent (SER)—Sphere+0.5 × Cylinder. Axial length (AL) was measured using non-contact partial coherence interferometry (Lenstar 900 instrument, Haag-Streit USA, Mason, Ohio, USA). VA is defined as habitual visual acuity, which was measured at 4 meters for each eye with participants' everyday refractive correction instruments (e.g., spectacles) using tumbling E Snellen visual charts. VA was recorded in logarithm of the minimum angle of resolution (LogMAR). SER, AL, and VA of the better refractive error eye of each participant was used for analysis. Presence of myopia macular degeneration (MMD) was defined as presenting at least one of 'diffuse chorioretinal atrophy', 'patchy chorioretinal atrophy', and 'macular atrophy' from colour fundus photograph grading, which was based on the International META-PM Classification [21]. The colour fundus photographs were captured for both eyes of the participants. All photographs were independently graded by two trained graders (LLW and ZQH), who were blinded to the participants' characteristics. In cases of disagreement during grading, a third trained grader (QVH) was consulted for adjudication.

### Definition of Visual Impairment (VI)

VI was classified based on habitual VA of the better eye (better SER). It was categorized as: none (VA < 0.30 LogMAR, or VA > 6/12 in Snellen), mild (0.30 LogMAR ≤ VA < 0.48 LogMAR, or 6/18 < VA ≤ 6/12 in Snellen), moderate (0.48

LogMAR ≤ VA < 1.00 LogMAR, or 6/60 < VA ≤ 6/18 in Snellen), and severe (VA ≥ 1.00 LogMAR, or VA ≤ 6/60 in Snellen). Because there was only one participant fall into the severe VI category, in the statistical analysis, this subject was grouped into the moderate VI category.

### Vision-related quality-of-life (VRQoL)

VRQoL was assessed using the 28-item C-IVI questionnaire [9]. This questionnaire contains one overall score and three domains: 'Mobility and independence' (MB), 'Reading and accessing information' (RD), and 'Emotional well-being' (EWB). The structure, item content, and domains are shown in S1 Table. Response options are three to four categories in Likert scale. In items 1–13, the responses options are: 'not at all' (0), 'a little' (1), 'quite a lot' (2), 'a lot' (3). In item 14–15, the responses options (numerical scores in brackets) are: 'not at all' (0), 'quite a lot' (1), 'a lot' (2). In item 16–28, the responses options are: 'never' (0), 'occasionally' (1), 'sometimes' (2), 'often' (3). Items 1–15 have an additional category: 'don't do this because of other reasons', which if selected, will remove from the analysis. During Rasch analysis (RA), these values were reversed so that higher logit scores indicate better VRQoL, and vice versa. In this study, the questionnaire was self-administered by the participants with instructions and explanations provided by the researchers.

### Rasch analysis (RA)

Rasch analysis was conducted using the Andrich rating scale model with Winsteps software (version 3.72.3, Chicago, Illinois, USA). To evaluate the psychometric properties of the C-IVI, we examined response category functioning, scale precision, unidimensionality, scale targeting, and differential item functioning [11,22]. The Rasch model assumes the probability of a respondent's choice in answering to an item is a logistic function of the relative distance of the item 'difficulty' level and the respondent's 'ability' level.

Category probability curves were used to visually inspect any 'Threshold disordering', which may indicate that a particular response category is underutilized. Sample-based person separation index (PSI), person reliability (PR), and range-based PR were used to determine scale precision, that is an indicator to show the instrument's ability to discriminate different levels of VRQoL. The minimally required scale precision to distinguish at least three levels of VRQoL are PSI > 2.0 and PR > 0.8 [11]. Because uncomplicated myopia is fully correctable optically, many participants registered maximum scores for most items; to account for this finding, range-based PR was also calculated [11]. To get the range-based PR, 'Statistically different levels of performance' (strata) was first manually calculated using the person measure standard errors generated during Rasch analysis. The range-based PR was then computed as $strata^2 \div (1 + strata^2)$. The number of performance strata was determined through the following steps: 1. begin at one end of the person measure (or raw score) range and proceed toward the other. 2. at each step, identify the next person measure that is separated from the current one by at least twice the joint standard error, where the joint standard error is the square root of the sum of the squared standard errors of the two measures (Joint SE = $sqrt(SE_1^2 + SE_2^2)$). 3. Continue this process, moving incrementally across the range, until no further statistically distinct steps can be identified. Each such step corresponds to one performance stratum [11,23]. The underlying assumption for each scale is unidimensionality, which is assessed using principal components analysis (PCA) of residuals. The raw variance explained by the first factor should exceed 50%, while the unexplained variance in the first contrast should be less than 2 eigenvalues [22]. The fit of each item within the corresponding scale is evaluated using the infit MnSQ statistic, which should be less than 1.3 [11]. Higher values indicate increased "noise" introduced by the item, compromising measurement accuracy. For instance, a value of 2 suggests 100% noise, meaning the noise is as significant as the signal. Scale targeting to respondents' level of the underlying VRQoL is examined through visual inspection of the Wright map. Bias in participant responses is tested using the differential item functioning (DIF).

## Statistical analysis

Descriptive statistics analyses were performed on participants' characteristics including sociodemographics, systemic diseases, and ocular characteristics. The unit of analysis was at patient level; in describing the ocular characteristics, the better spherical equivalent eye was chosen for analysis. Continuous variables were summarized by mean and standard deviation (SD), and categorical variables were described by frequencies and proportions.

In the criterion validity assessment, to show the C-IVI instruments can discriminate participants of different severity of VI, analysis of variance (ANOVA) test and post-hoc pairwise t-test were used to compare Rasch-transformed scores in the 1) none VI, 2) mild VI, 3) moderate VI groups (one severe VI participant was combined to moderate VI group). Since the participants of this study were all highly myopic adults, additionally, we examined the relationship between the VRQoL and SER, and the relationship between the VRQoL and presence of MMD. The analysis was the same as criterion validity assessment, using ANOVA and post-hoc pairwise t-test. SER was categorized in four categories based on the less myopic eye of each patient: 1) SER ≥ −7.00D, 2) −10.00D ≤ SER < −7.00D, 3) −15.00D ≤ SER < −10.00D, 4) SER < −15.00D. All statistical analyses were conducted using SAS 9.4 (SAS Institute, Carry, NC, USA). Two-sided $P < 0.05$ was considered statistically significant.

## Results

### Participants' characteristics

A total of 431 participants were included in this study. Patients' characteristics are summarized in Table 1. The mean (SD) of age of the participants was 42.2 (7.1) years, with 65.4% being female. MMD was present in 15.8% of participants. The mean (SD) VA was 0.1 (0.2) LogMAR, with 79.4%, 13.5%, 7.0%, and 0.2% showing none, mild, moderate and severe VI, respectively.

### Response category functioning and Scale precision

Overall score: The thresholds of the 28-item overall C-IVI and the three domains were well-ordered, as illustrated in Fig 1. The category probability curves confirmed that response categories were appropriately structured based on item difficulty, with the highest probability corresponding to the appropriate difficulty level. As shown in Table 2, PSI and PR of the overall C-IVI were suboptimal, with values of 1.82 and 0.77, lower than the 2.0 and 0.8 cutoff, respectively. However, the range-based PR was excellent, 0.94 for the overall C-IVI. We found items 14, 15, 25, 27 were misfitting, the infit MnSq were 1.65, 1.33, 1.35, 1.53. Sample-based PSI (1.70) and PR (0.74) did not improve after the removal of the misfit items. Since items 14 and 15 represent essential daily activities commonly performed by our participants, and items 24 and 27 capture important aspects of emotional well-being relevant to adults with high myopia, these items were retained in the instrument.

Domain Scores: The range-based PR was 0.80 in all three domains. Item 6 (infit MnSq = 1.31), 14 (infit MnSq = 1.48), 15 (infit MnSq = 1.35) were misfitting in the MB domain. Item 19 (infit MnSq = 1.46) was misfitting in the RD domain. Item 27 (infit MnSq = 1.37) was misfitting in the EWB domain. For the same reason as the overall C-IVI, none of the misfit items were removed. The suboptimal precision was expected in this community-based sample, as 79.4% of the participants presented with no VI, and the ceiling effect was quite large (17.6%, 35.5%, 41.3%, and 31.6% in overall and three domains, respectively, shown in Table 2. Item Frequency table was shown in S2 Table.

### Unidimensionality and targeting

Multidimensionality was shown in the overall C-IVI (Table 2). The variance explained by the 1st factor was 47.3%, and the eigenvalue for the first contrast was 3.5. However, in the three domains, each was unidimensional, and had eigenvalue for the first contrast lower than 2: 1.8 for the MB, 1.8 for the RD, and 1.7 for the EWB. These statistics support the split of the

**Table 1. Patient characteristics of the 431 participants from high myopia adult cohort in Changsha, central China.**

| Characteristics | Overall |
|---|---|
| Age (Years)* | 42.2 (7.1) |
| Age (Categorical) | |
| ≤40 | 200 (46.4%) |
| >40 | 231 (53.6%) |
| Gender | |
| Male | 149 (34.6%) |
| Female | 282 (65.4%) |
| Ethnicity | |
| Han | 410 (95.1%) |
| Others | 21 (4.9%) |
| Education level | |
| high school or below | 44 (10.2%) |
| college or above | 387 (89.8%) |
| Diabetes | |
| No | 424 (98.4%) |
| Yes | 7 (1.6%) |
| Hypertension | |
| No | 407 (94.4%) |
| Yes | 24 (5.6%) |
| presence of MMD | |
| No | 363 (84.2%) |
| Yes | 68 (15.8%) |
| Axial length (mm)* | 26.8 (1.7) |
| Axial length (Categorical) | |
| <26.9 | 257 (59.6%) |
| ≥26.9 | 174 (40.4%) |
| Sphereical Equivalent (D)* | −8.3 (3.8) |
| Sphereical Equivalent (Categorical) | |
| ≥−7.00 | 206 (47.8%) |
| -10.00 to −7.00 | 130 (30.2%) |
| -15.00 to −10.00 | 70 (16.2%) |
| <−15.00 | 25 (5.8%) |
| Habitual visual acuity of the better eye (LogMAR)* | 0.1 (0.2) |
| Vision impairment | |
| None | 342 (79.4%) |
| Mild | 58 (13.5%) |
| Moderate | 30 (7.0%) |
| Severe | 1 (0.2%) |

*Continous measures were reported as mean (standard deviation)

C-IVI into the above three domains. Targeting of the overall C-IVI and its three domains was assessed in Fig 2, showing differences between person and item means of 3.81 logits, 3.89 logits, 5.20 logits, and 3.64 logits, respectively. These findings indicate that the participants were too able for the items.

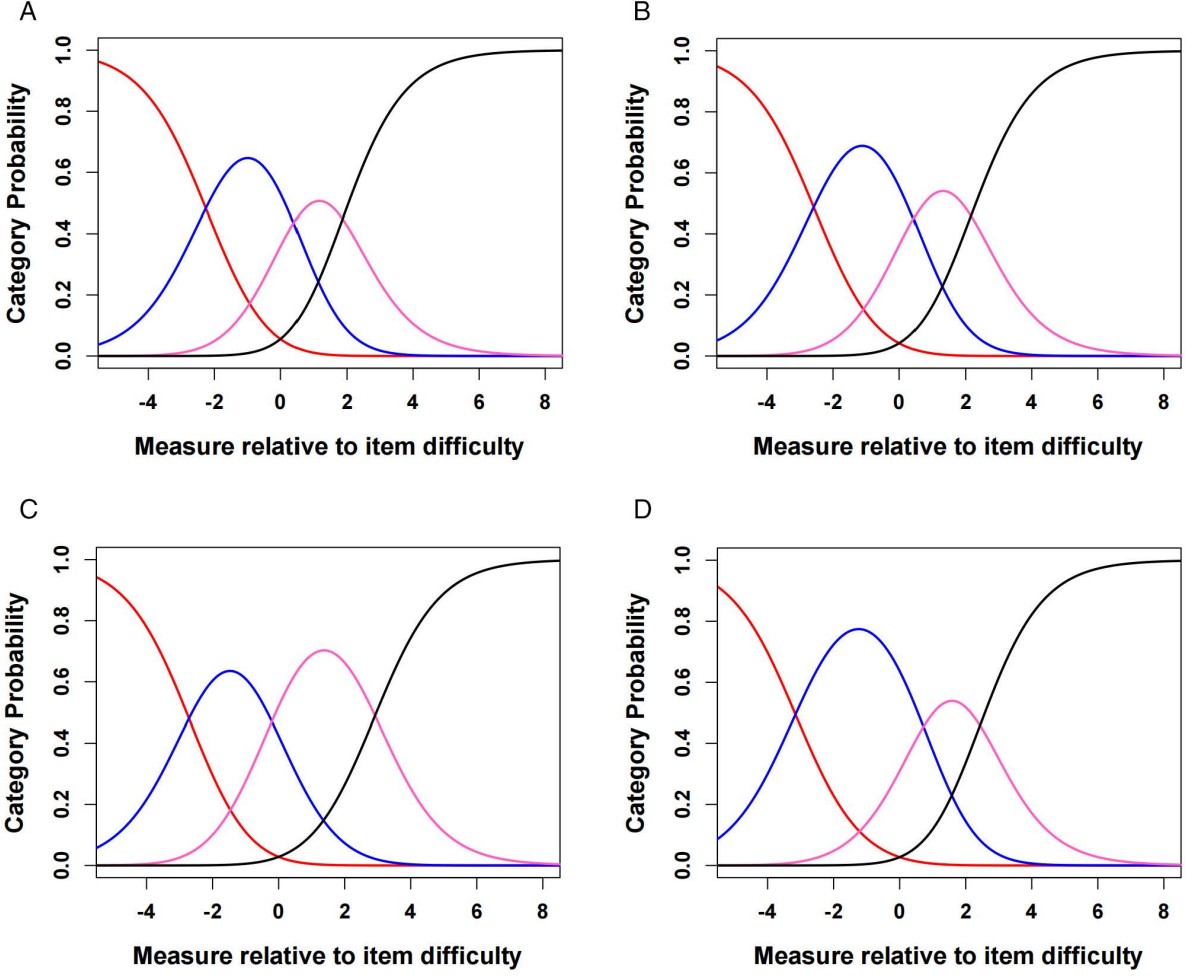

**Fig 1. Category probability curves for the C-IVI showed ordered thresholds.** A. in the 28-item C-IVI; B. in the 'Mobility and independence' domain; C. in the 'Reading and accessing information' domain; D. in the 'Emotional well-being' domain. Red, blue, pink, black color lines represents the 'not at all' (0), 'a little' (1), 'quite a lot' (2), 'a lot' (3) categories of the items.

## Differential item functioning (DIF)

The DIF was uniform for the overall C-IVI and the EWB domain. DIF for age was found in item 6 in the MB domain. DIF for gender was found in item 6 in the MB domain, and item 2 in the RD domain. We further revised the MB domain without item 6, and the RD domain without item 2. The revised domains showed uniform DIF in age and gender (Table 2). The person scores of these two domains were output from the revised domains.

## Criterion validity

As shown in Table 3, the Rasch transformed scores decrease with severity of VI. In the Overall C-IVI, the mean (SD) of the score was 4.0 (1.9) logits among None VI, 3.2 (1.8) logits among Mild VI, and 2.6 (2.0) logits among Moderate VI (one Severe VI participant was combined to Moderate VI), with $p < 0.001$ from ANOVA test. The same trend was observed in all three domains. In pair-wise comparisons, only p-values from adjacent groups were reported, results were significant for all pairs, except mild VI vs. moderate VI in the overall C-IVI (3.2 (1.8) vs. 2.6 (2.0), $p = 0.11$), and the EWB domain (3.0 (2.3)

**Table 2. Fit parameters of the C-IVI and its three component domains compared to Rasch model (N = 431).**

| Parameters | Rasch Model | C-IVI 28 | Mobility and independence | Mobility and independence: Revised | Reading and accessing information | Reading and accessing information: Revised | Emotional well-being |
|---|---|---|---|---|---|---|---|
| Items | | 1-28 | 1,3,5-9,14,15 | 1,3,5,7-9,14,15 | 2,4,10-13,16-20 | 4,10-13,16-20 | 21-28 |
| Disordered thresholds | No | No | No | No | No | No | No |
| Person separation index (sample-based) | >2.0 | 1.82 | 0.97 | 0.90 | 1.07 | 0.98 | 1.47 |
| Person reliability (sample-based) | >0.8 | 0.77 | 0.49 | 0.45 | 0.53 | 0.49 | 0.68 |
| Person reliability (range-based) | Strata, PR | 4, 0.94 | 2, 0.80 | 2, 0.80 | 2, 0.80 | 2, 0.80 | 2, 0.80 |
| Item fit (infit MnSq) | <1.3 | **Item 14(1.65) Item 15(1.33) Item 25(1.35) Item 27(1.53)** | **Item 6(1.31) Item 14(1.48) Item 15(1.35)** | **Item 14(1.43) Item 15(1.32)** | **Item 19(1.46)** | **Item 19(1.50)** | **Item 27(1.37)** |
| PSI after removal misfit items (sample-based) | >2.0 | 1.70 | 1.01 | 1.01 | 1.05 | 0.97 | 1.51 |
| PR after removal misfit items (sample-based) | >0.8 | 0.74 | 0.51 | 0.51 | 0.52 | 0.48 | 0.70 |
| PCA:Variance by the 1st factor | >50% | 47.3% | 46.2% | 48.3% | 50.1% | 50.8% | 55.5% |
| PCA:Eigenvalue for 1st contrast | <2.0 | 3.5 | 1.8 | 1.8 | 1.8 | 1.8 | 1.7 |
| Targeting, difference between person and item mean | <1.0 | 3.81 | 3.89 | 4.08 | 5.20 | 5.38 | 3.64 |
| Ceiling effect (extreme maximum scores; n,%) | – | 76, 17.6% | 153, 35.5% | 179, 41.5% | 178, 41.3% | 190, 44.1% | 136, 31.6% |
| DIF* | <1.0, p>0.05 | None | **Age (Item6, DIF contrast = 1.55,p < 0.001) Gender (Item6, DIF contrast = 1.08,p < 0.001)** | None | **Gender (Item2, DIF contrast = 1.01,p = 0.002)** | None | None |

*DIF was tested for age group (<40 years, ≥ 40 years), and gender (male, female)

vs. 3.0 (2.8), p = 0.97). Habitual VA in the better eye was moderately correlated with the overall C-IVI score (r = −0.27) and its three domains (r = −0.34, −0.32, and −0.16, respectively). SER in the better eye showed positive correlations with the overall C-IVI score (r = 0.33) and with each domain (r = 0.30, 0.28, and 0.32, respectively).

The C-IVI instrument also showed commendable discrimination of SER and MMD. However, in the overall C-IVI and three domains, the difference between '-15.00D≤SER<-10.00D' and 'SER<-15.00D' were not statistically significant. And the difference between 'SER≥-7.00D' and '-10.00D≤ SER<-7.00D' were not significant in the 'Mobility and independence' and 'Reading and accessing information' domains. In the criterion validity of MMD, the mean (SD) of overall C-IVI scores was 2.7 (1.9) logits in the MMD and 4.0 (1.9) logits in the non-MMD (p < 0.001), the MB domain was 3.0 (2.0) logits in the MMD and 4.3 (1.7) logits in the non-MMD (p < 0.001), the RD domain was 2.7 (1.9) logits in the MMD and 4.0 (1.9) logits in the non-MMD (p < 0.001), and the EWB domain was 2.4 (2.5) logits in the MMD and 3.9 (2.1) logits in the non-MMD (p < 0.001).

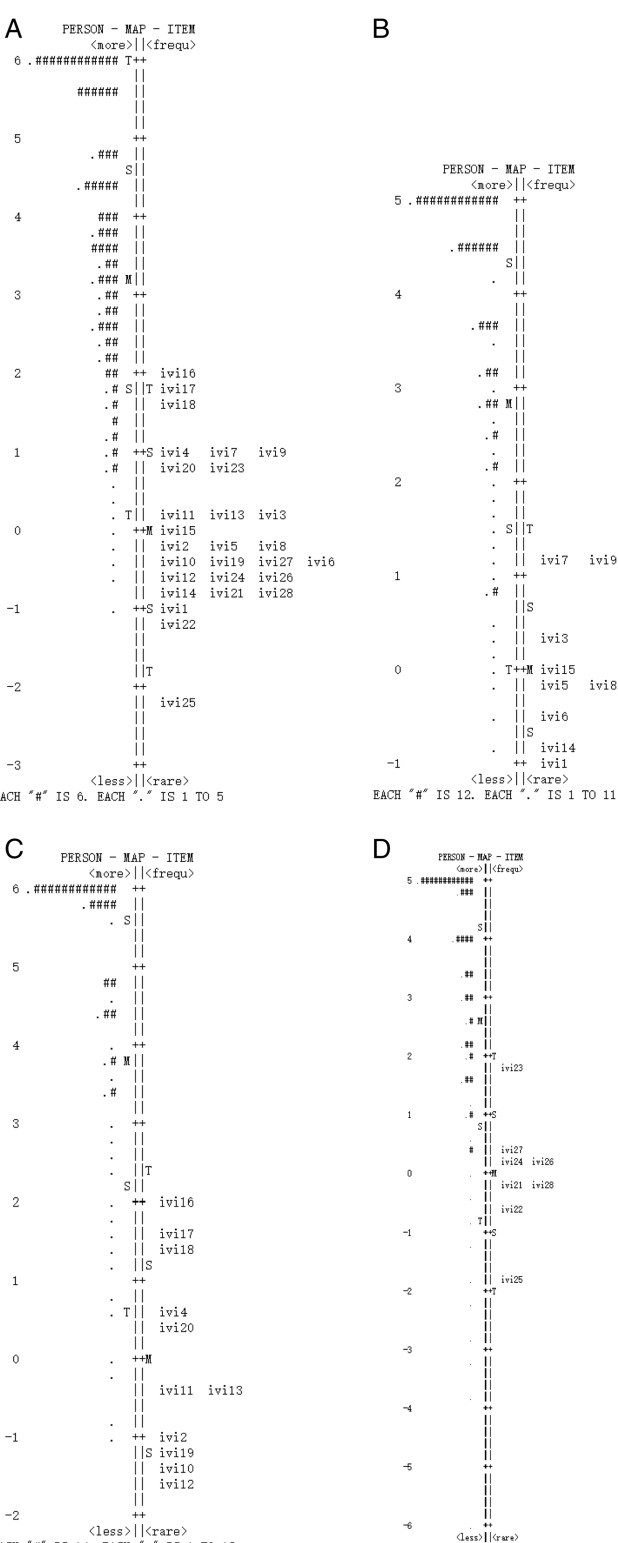

**Fig 2. Person-Item map for the 28-item C-IVI.** Left of the dashed line represents participants. Right of the dashed line represents items, denoted by their item number. Top of the diagram are participants with better VRQoL and less difficult items. Bottom of the diagram are participants with worse VRQoL and more difficult items. A. in the 28-item C-IVI, '#' signifying 6 participants, and '.' signifying 1-5 participants; B. in the 'Mobility and

independence' domain, '#' signifying 12 participants, and '.' signifying 1-11 participants; C. in the 'Reading and accessing information' domain, '#' signifying 14 participants, and '.' signifying 1-13 participants; D. in the 'Emotional well-being' domain, '#' signifying 11 participants, and '.' signifying 1-10 participants.

## Discussion

In this study, we assessed the psychometric properties of the 28-item C-IVI in adults with high myopia in China. Our findings suggest that the overall C-IVI and its three domains, 'Mobility and Independence,' 'Reading and Accessing Information,' and 'Emotional Well-being'—produce reliable person estimates, despite a large proportion of participants not exhibiting VI. The precision was suboptimal, and targeting was poor, likely due to the sample being too 'able'. Participants were highly myopic adults, yet 79.4% were not visually impaired based on habitual VA in the better eye. They were also relatively young, with a mean age of 42.2 years, and 89.8% had attained a college-level education or higher. These characteristics resulted in limited variance within the sample, with many participants reaching the ceiling effect. Nevertheless, our findings demonstrated commendable discrimination of VI, spherical equivalent (SER), and myopic macular

Table 3. Criterion validity of the C-IVI questionnaire.

| | C-IVI 28 | Mobility and independence | Reading and accessing information | Emotional well-being |
|---|---|---|---|---|
| | Range: −0.9, 6.8 | Range: −0.8, 5.8 | Range: −1.2, 7.1 | Range: −6.8, 6.1 |
| Vision Impairment (VI) | | | | |
| a) None (N = 342) | 4.0 (1.9) | 4.3 (1.7) | 5.6 (1.8) | 3.8 (2.1) |
| b) Mild (N = 58) | 3.2 (1.8) | 3.6 (1.9) | 4.8 (2.0) | 3.0 (2.3) |
| c) Moderate (N = 31)* | 2.6 (2.0) | 2.3 (1.9) | 3.7 (2.3) | 3.0 (2.8) |
| P-value of ANOVA | <0.001 | <0.001 | <0.001 | 0.02 |
| P-value of a vs. b | 0.004 | 0.004 | <0.001 | 0.01 |
| P-value of b vs. c | 0.11 | 0.003 | 0.02 | 0.97 |
| Spearman Correlation between habitual VA and C-IVI | −0.27 | −0.34 | −0.32 | −0.16 |
| Spherical Equivalent (SER) | | | | |
| a) SER ≥ −7.00D (N = 206) | 4.3 (1.8) | 4.4 (1.6) | 5.8 (1.7) | 4.2 (2.0) |
| b) −10.00D ≤ SER < −7.00D (N = 130) | 3.9 (1.8) | 4.3 (1.7) | 5.5 (1.8) | 3.7 (2.0) |
| c) −15.00D ≤ SER < −10.00D (N = 70) | 2.8 (1.7) | 3.2 (1.9) | 4.5 (2.3) | 2.6 (2.3) |
| d) SER < −15.00D (N = 25) | 2.2 (1.8) | 2.6 (2.1) | 3.9 (2.5) | 1.6 (2.7) |
| P-value of ANOVA | <0.001 | <0.001 | <0.001 | <0.001 |
| P-value of a vs. b | 0.02 | 0.41 | 0.19 | 0.04 |
| P-value of b vs. c | <0.001 | <0.001 | <0.001 | <0.001 |
| P-value of c vs. d | 0.14 | 0.17 | 0.30 | 0.09 |
| Spearman Correlation between SER and C-IVI | 0.33 | 0.30 | 0.28 | 0.32 |
| Presence of Myopia Macular Degeneration (MMD) | | | | |
| No (N = 363) | 4.0 (1.9) | 4.3 (1.7) | 5.6 (1.8) | 3.9 (2.1) |
| Yes (N = 68) | 2.7 (1.9) | 3.0 (2.0) | 4.5 (2.4) | 2.4 (2.5) |
| P-value | <0.001 | <0.001 | <0.001 | <0.001 |

Measurements reported by Mean (SD) in logits, higher logit scores indicate better VRQoL

VI, SER, and MMD were based on the less myopic eye of each patient

*1 participant with Severe VI was combined to Moderate VI

degeneration (MMD) based on person scores. High myopia adults with MMD showed lower VRQoL than those without MMD with statistical significance.

To the best of our knowledge, this is the first validation of the 28-item C-IVI questionnaire in a cohort with high myopia. The C-IVI was previously validated in a population-based study in Singapore using the 32-item questionnaire [11]. Fenwick and colleagues reported similar findings to our study, with sample-based precision at 0.69, improving to 0.99 for range-based precision. This was attributed to a large proportion of participants being too 'able' (72.8% without VI and 55.4% at ceiling). Interestingly, in both studies, criterion validity assessments revealed non-significant differences in person scores between mild and moderate VI groups in the EWB domain. However, direct comparisons between studies are challenging, as Rasch-transformed scores are relative measures specific to each study.

One of the study critical findings was that the decline in VRQoL with worsening SER or the presence of MMD was consistent with increasing VI severity. High myopia has been reported to increase the risk of MMD, with odds ratios of 62.3 and 11.6 compared to low and moderate myopia [3]. Furthermore, higher severity of myopia is typically defined by worse SER. In our study, worse SER and the presence of MMD were associated with more severe VI. Clinically, stronger corrective lenses are required for worse SER, often resulting in thicker spectacles. However, several effective methods exist to correct high and even very high myopia (SER < −15.00D) (e.g., LASIK surgeries, implantable collamer lenses (ICL), rigid gas permeable lenses). Unfortunately, our study cannot determine whether participants with moderate or severe VI had irreversible vision loss or if they were under-corrected due to economic or other considerations. Future research should explore the reasons for under-correction in highly myopic patients and its impact on their mental health and well-being.

This study has several strengths, including its relatively large sample size, community-based design, focus on highly myopic adults, use of modern psychometric methods, and inclusion of a range-based reliability coefficient. However, there are also limitations. First, the sample underrepresented individuals with severe VI, limiting the generalizability of our findings. Nevertheless, we observed a clear trend of declining VRQoL with increasing VI severity. Second, precision and targeting were suboptimal, which was expected given the high proportion of participants without VI. These properties are likely to be improved in future studies with a greater representation of visually impaired individuals. Third, we did not adjust pairwise comparisons for multiple comparisons, as the groups were categorized into three or four levels, and there was a clear trend in mean value changes.

In summary, our study demonstrated that the 28-item C-IVI questionnaire provides reliable person estimates of VRQoL in high myopia adult cohort in China. A higher degree of myopia and the presence of MMD, consistent with worse VI, were associated with lower VRQoL scores across all three domains. Future studies should include more visually impaired individuals to further validate the C-IVI questionnaire and assess its applicability in diverse populations.

## Supporting information

**S1 Table: Structure, Item Content, and Domain of the C-IVI questionnaire.**
(DOCX)

**S2 Table: Item Frequency Table of the C-IVI 28.**
(DOCX)

**S3 File: Chinese Impact of Vision Impairment (C-IVI) questionnaire in Chinese and English.**
(DOCX)

## Author contributions

**Conceptualization:** Ecosse L. Lamoureux, Weizhong Lan.

**Data curation:** Ziqi Hu, Yanfeng Jiang, Huanhuan Tan, Zhongping Chen.

**Formal analysis:** Wei Pan.

**Funding acquisition:** Weizhong Lan.

**Investigation:** Weizhong Lan.

**Methodology:** Wei Pan, Ryan E K Man, Eva K Fenwick, Quan V. Hoang, Seang-Mei Saw, Ecosse L. Lamoureux, Weizhong Lan.

**Project administration:** Weizhong Lan.

**Supervision:** Weizhong Lan, Zhikuan Yang.

**Writing – original draft:** Wei Pan.

**Writing – review & editing:** Wei Pan, Ryan E K Man, Eva K Fenwick, Seang-Mei Saw, Ecosse L. Lamoureux, Weizhong Lan, Zhikuan Yang.

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
