## [Decision Letter · Decision Letter 0]

23 Jul 2025

PONE-D-25-33208A psychometric evaluation of the Chinese Impact of Vision Impairment (C-IVI) questionnaire in an adult cohort with high myopia using Rasch AnalysisPLOS ONE

Dear Dr. Lan,

Thank you for submitting your manuscript to PLOS ONE. After careful consideration, we feel that it has merit but does not fully meet PLOS ONE’s publication criteria as it currently stands. Therefore, we invite you to submit a revised version of the manuscript that addresses the points raised during the review process.

We look forward to receiving your revised manuscript.

Kind regards,

Andrzej Grzybowski

Academic Editor

PLOS ONE

Journal Requirements:

4. In the online submission form, you indicated that the data that support the findings of this study are not openly available due to reasons of confidentiality. Upon reasonable request, deidentified data can be accessed. Please contact corresponding author, Prof. Weizhong Lan (lanweizhong@aierchina.com).

5. Please amend your authorship list in your manuscript file to include author Quan Hoang, Man Ek.

6. Please amend the manuscript submission data (via Edit Submission) to include author Man Ryan EK, Quan V. Hoang.

7. Please amend your list of authors on the manuscript to ensure that each author is linked to an affiliation. Authors’ affiliations should reflect the institution where the work was done (if authors moved subsequently, you can also list the new affiliation stating “current affiliation:….” as necessary).

Reviewers' comments:

Reviewer's Responses to Questions

**Comments to the Author**

1. Is the manuscript technically sound, and do the data support the conclusions?

Reviewer #1: Yes

Reviewer #2: Yes

2. Has the statistical analysis been performed appropriately and rigorously? 

Reviewer #1: Yes

Reviewer #2: Yes

3. Have the authors made all data underlying the findings in their manuscript fully available?

Reviewer #1: Yes

Reviewer #2: No

4. Is the manuscript presented in an intelligible fashion and written in standard English?

Reviewer #1: Yes

Reviewer #2: Yes

5. Review Comments to the Author

Reviewer #1: The paper reported the findings from a psychometric evaluation of the Chinese Impact of Vision Impairment (C-IVI) questionnaire in a large group of patients with high myopia using Rasch Analysis. The study was well-conducted overall. The paper clearly presented some interesting results. Some comments are provided to improve the paper.

1. Line 58, please change “VA” to “uncorrected VA”.

2. Lines 196-197, please indicate that SER from less myopic eye of a patient was used to define SER groups.

3. In Method, please mention where these 445 patients with high myopia were enrolled from.

4. Table 1, please footnote that the results for continuous measures (e.g., axial length) were reported as mean (SD).

5. Table 1, please indicate the SER reported was from less myopic eye. Why using -7.9 D as the cutoff when categorizing SER into two groups? It may be better to use the clinically meaningful cutoff for categorizing SER into groups.

6. In table 1, please indicate VA as “Habitual visual acuity”.

7. Figure 1 legend, please indicate what each line of different color represents for.

8. Figure 2 legend , please change “fr” to “for”.

9. eTable 2, are the SER, VI, and MMD all from the same eye (i.e., the eye with less myopic SER)? Please add a footnote to clarify this.

10. Table 3, it will be good to indicate the possible range of overall sore of C-IVI, and each domain score, and indicate higher score means better vision function.

11. Besides the analysis categorizing VA, SER into levels, it may be worthy evaluating the correlation of habitual visual acuity and SER (analyzed as continuous measures) with C-IVI overall score and each domain scores. This correlation analysis may be performed based on both worse eye (more myopic eye) and better eye (less myopic eye), because it is possible that correlations are higher using worse eye than using the better eye.

12. Lines 294 to 295, it stated that high myopia has been reported to increase the risk of MMD, with an odds ratio of 845 compared to non-myopia”, since MMD is only specific to myopic eye, it does not make sense to compare it to non-myopic eye. Did you mean “compared to non-high myopic eye”?

13. Line 297, what is “vsl”?

Reviewer #2: This study reports the psychometric validation of the Chinese Impact of Vision Impairment (C-IVI) questionnaire in adults with high myopia. Please see some queries and suggestions below to improve the clarity of the manuscript.

Given that Rasch analysis can handle missing data, what was the rationale for excluding incomplete C-IVI responses? What percentage of responses were missing? Was the missingness random or systematic?

Can you provide the item frequency tables to report ceiling effects?

The authors note that range-based PR was used due to ceiling effects. Please describe how range-based PR is calculated and its practical relevance compared to sample-based PR?

Given that the scales had low sample-based PR, how might this impact the accuracy of person estimates and the validity of comparisons between sub-groups?

One of the aims was to examine relationship between VRQoL and myopia macular degeneration (MMD). However, the sample included only 68 individuals with MMD. The IVI was designed for people with vision impairment; so, it is not surprising that the items are less difficult than the abilities of the sample, who predominantly had no VI (n=342). Should more data be collected, especially from individuals with MMD, for more reliable validation?

If someone wants to use the C-IVI in a similar adult population with myopia, can they use the item and category calibrations from this study to derive person estimates? What limitations should be considered? It would be helpful to be transparent about further application in the discussion.

Line 197-8: ‘the relationship between VI and SE, as well as VI and MMD were examined utilizing chi-squared test.’ It is not clear how this aligns with the research questions.

Minor:

Abstract: ‘SER’ in results was not expanded before.

Can the subheadings be written in full rather than abbreviations (e.g. VRQoL, RA)?

Line 157: The description under Rasch analysis needs revision. “Andrich rating scale model was used to analyse the C-IVI data in the RA” is not clear.

Line 158: “To assess the psychometric properties of the C-IVI, response category functioning, scale precision, unidimensionality, scale targeting, and differential item functioning” – this sentence is not complete.

Line 166 – “person readability” should be ‘person reliability’

Line 192: 1 participant with severe VI was combined with moderate VI group for analysis. However, the eTable 2 corresponding line in text (line 208-209) reports severe VI group instead of moderate.

Line 297: what is ‘Vsl’?

eTable 1: item 18, ‘fafety’ should be ‘safety’

6. PLOS authors have the option to publish the peer review history of their article (what does this mean? ). If published, this will include your full peer review and any attached files.

**Do you want your identity to be public for this peer review?** For information about this choice, including consent withdrawal, please see our Privacy Policy .

Reviewer #1: No

Reviewer #2: No

---

## [Author Response · Author response to Decision Letter 1]

18 Aug 2025

Dear Editor and anonymous reviewers,

Many thanks for your insightful and constructive comments and suggestions. We have made responses point-by-point and revised the manuscript accordingly. With your help, we believe the quality of the manuscript has improved significantly and hope this updated version would meet the standard of the journal.

Thank you for your consideration and looking forward to your feedback.

Sincerely yours,

Weizhong Lan

MD, PhD

Professor of Ophthalmology

Central South University, China

Reviewer #1: The paper reported the findings from a psychometric evaluation of the Chinese Impact of Vision Impairment (C-IVI) questionnaire in a large group of patients with high myopia using Rasch Analysis. The study was well-conducted overall. The paper clearly presented some interesting results. Some comments are provided to improve the paper.

1.Line 58, please change “VA” to “uncorrected VA”.

Reply: Thank you for this comment, we have revised “VA” to “uncorrected VA” as suggested.

2.Lines 196-197, please indicate that SER from less myopic eye of a patient was used to define SER groups.

Reply: Thank you for your suggestion. We have indicated that the SER was from less myopic eye of each patient for clarity.

3.In Method, please mention where these 445 patients with high myopia were enrolled from.

Reply: We have added the following to the Methods section: “A total of 445 participants were enrolled at the Changsha Aier Eye Hospital in Hunan Province, China, through social media outreach and word-of-mouth referrals within the community.” in Line 101-103.

4.Table 1, please footnote that the results for continuous measures (e.g., axial length) were reported as mean (SD).

Reply: Thank you. We have added the footnote specify that contentious measures were reported in mean (standard deviation).

5.Table 1, please indicate the SER reported was from less myopic eye. Why using -7.9 D as the cutoff when categorizing SER into two groups? It may be better to use the clinically meaningful cutoff for categorizing SER into groups.

Reply: We have indicated that SER was from the less myopic eye. Regarding categorization, although the original cutoff of -7.9 D was based on the median reported in our baseline paper of this cohort, we agree with your suggestion and have revised the categories to clinically meaningful groups: 1) SER≥-7.00D, 2) -10.00D≤SER<-7.00D, 3) -15.00D≤SER<-10.00D, 4) SER<-15.00D.

6.In table 1, please indicate VA as “Habitual visual acuity”.

Reply: Thank you for the comment, we have updated the table to indicate “habitual visual acuity” as suggested.

7.Figure 1 legend, please indicate what each line of different color represents for.

Reply: Thank you, this is a very good point. We have added the explanation in the legend as suggested.

8.Figure 2 legend , please change “fr” to “for”.

Reply: Thank you. We have corrected the spelling, and re-examined the manuscript for similar mistakes.

9.eTable 2, are the SER, VI, and MMD all from the same eye (i.e., the eye with less myopic SER)? Please add a footnote to clarify this.

Reply: Thank you for the comments, yes, the SER, VI, and MMD all from the less myopic eye. Due to redundant information to Table 3, we have decided to remove eTable 2. For Table 3 footnote, we added the clarification as suggested.

10.Table 3, it will be good to indicate the possible range of overall sore of C-IVI, and each domain score, and indicate higher score means better vision function.

Reply: Thank you. We have added a row to expand the range of transformed scores of C-IVI, and each domain.

11.Besides the analysis categorizing VA, SER into levels, it may be worthy evaluating the correlation of habitual visual acuity and SER (analyzed as continuous measures) with C-IVI overall score and each domain scores. This correlation analysis may be performed based on both worse eye (more myopic eye) and better eye (less myopic eye), because it is possible that correlations are higher using worse eye than using the better eye.

Reply: Thank you for the valuable suggestion. We have added the correlation analyses between habitual visual acuity (VA), spherical equivalent refraction (SER), and C-IVI scores in Table 3 and the main text (Line 267-270). The correlation coefficients were very similar between the better and worse eye. To maintain consistency with the rest of the analysis, we reported only the results from the better eye.

Specifically, the correlation coefficients between VA in the better eye and the C-IVI overall score and its three domains were -0.27, -0.34, -0.32, and -0.16, respectively. For the worse eye, the corresponding values were -0.28, -0.32, -0.31, and -0.20.

Similarly, the correlations between SER in the better eye and the C-IVI overall score and its domains were 0.33, 0.30, 0.28, and 0.32, respectively; and for the worse eye, 0.31, 0.29, 0.26, and 0.29.

12.Lines 294 to 295, it stated that high myopia has been reported to increase the risk of MMD, with an odds ratio of 845 compared to non-myopia”, since MMD is only specific to myopic eye, it does not make sense to compare it to non-myopic eye. Did you mean “compared to non-high myopic eye”?

Reply: Thank you for the comment. We have changed the sentence to “High myopia has been reported to increase the risk of MMD, with odds ratios of 62.3 and 11.6 compared to low-myopia and moderate myopia” in Line 310-311.

13. Line 297, what is “vsl”?

Reply: Thank you. We have corrected the typo ‘VsI' to ‘VI'.

Reviewer #2: This study reports the psychometric validation of the Chinese Impact of Vision Impairment (C-IVI) questionnaire in adults with high myopia. Please see some queries and suggestions below to improve the clarity of the manuscript.

Given that Rasch analysis can handle missing data, what was the rationale for excluding incomplete C-IVI responses? What percentage of responses were missing? Was the missingness random or systematic?

Reply: Thank you for the comment. A total of 14 out of 445 participants (approximately 3.1%) did not complete the C-IVI questionnaire (without any data entry submitted) at baseline. Given the small proportion, we chose to exclude these subjects for analysis. While Rasch analysis can indeed handle missing data within completed questionnaires, our exclusion was due to the questionnaires being entirely or substantially incomplete.

The primary reasons for non-completion were logistical rather than systematic. Participants often had a heavy testing schedule and were asked to complete the questionnaire intermittently between clinical examinations. Some began filling out the C-IVI during pupil dilation but were unable to finish due to blurred vision. Although a few indicated they would complete it at home, the questionnaires were ultimately not returned. Given these circumstances, the missingness was minimal and appears to be random or due to logistical constraints, rather than related to participants' vision or other systematic factors.

Can you provide the item frequency tables to report ceiling effects?

Reply: Thank you. We have added the item frequency table of C-IVI (28 items) in eTable 3.

The authors note that range-based PR was used due to ceiling effects. Please describe how range-based PR is calculated and its practical relevance compared to sample-based PR?

Reply: Thank you for this important comment. Sample-based person reliability (PR) reflects how well the instrument differentiates among specific individuals within the study sample. It is calculated based on the observed variance in person measures. In situations where the sample has little variability (e.g., most participants score at the ceiling), the reliability would appear artificially low, even if the instrument itself was capable of distinguishing performance levels in a broader population.

In our study, most of our participants did not have vision impairment, despite the instrument (IVI) being designed for visually impaired individuals. This led to ceiling effects, where many participants scored at the upper end of the scale, reducing the observed spread of person measures and thus producing a low sample-based PR.

To address this limitation, we used range-based PR, which instead considers the expected measurement precision across the full range of the scale by using standard errors of person estimates. This method estimates how many statistically distinguishable performance strata the scale can detect (independent of the current sample's distribution). It provides a better reflection of the scale’s theoretical discriminating power, especially when applied to a restricted or skewed sample.

The process was previously described in “Wright, B. (2001). Separation, reliability and skewed distributions: Statistically different levels of performance. Rasch Measurement Transactions, 14(4), 786”. “Statistically different levels of performance” (strata) were manually calculated using the person measure standard errors generated during Rasch analysis, and the reliability is equal to strata2 ÷ ( 1+ strata2 ). The calculation steps for strata are: 1. begin at one end of the person measure (or raw score) range and proceed toward the other. 2. at each step, identify the next person measure that is separated from the current one by at least twice the joint standard error, where the joint standard error is the square root of the sum of the squared standard errors of the two measures (Joint SE = sqrt(SE12+SE22)). 3. Continue this process, moving incrementally across the range, until no further statistically distinct steps can be identified. Each such step corresponds to one performance stratum.

We now had merit this calculation process into the manuscript, Lines 173-183.

Given that the scales had low sample-based PR, how might this impact the accuracy of person estimates and the validity of comparisons between sub-groups?

Reply: We appreciate this thoughtful comment. Indeed, low sample-based PR implies limited ability to differentiate between individuals within this specific sample. Differences between sub-groups should be interpreted with caution, particularly when these sub-groups also exhibit low variability in their responses. However, it is important to note that Rasch person estimates remain mathematically valid and interval-scaled, even under low reliability.

One of the aims was to examine relationship between VRQoL and myopia macular degeneration (MMD). However, the sample included only 68 individuals with MMD. The IVI was designed for people with vision impairment; so, it is not surprising that the items are less difficult than the abilities of the sample, who predominantly had no VI (n=342). Should more data be collected, especially from individuals with MMD, for more reliable validation?

Reply: Thank you for this observation. The main goal of the manuscript is to demonstrate that the C-IVI questionnaire is valid for use in a high myopia cohort. We agree with the reviewer that including a larger proportion of individuals with MMD could provide a more specific assessment of the relationship between VRQoL and MMD. In our sample, the prevalence of MMD was approximately 15.8% (N = 68); however, this still provided sufficient power to detect differences in VRQoL between participants with and without MMD, as shown in Table 3.

If someone wants to use the C-IVI in a similar adult population with myopia, can they use the item and category calibrations from this study to derive person estimates? What limitations should be considered? It would be helpful to be transparent about further application in the discussion.

Reply: Yes, we believe the C-IVI can also applied to similar adult population with myopia. Our study showed validity of C-IVI in high myopia cohort. Given that myopia covers a broad spectrum of clinical and functional levels, there may be differences in the item and category calibrations in other populations with myopia depending on the severity level, type of correction used, and type of complications that participants present with, particularly if the sample population in question is small. There are certain limitations need to be noted: 1) items may not optimally targeted to the ability range of sample, which may reduce measurement precision. 2) the C-IVI was originally designed for populations with vision impairment. Its sensitivity and relevance in high-functioning or asymptomatic myopia patients may be limited. 3) the small number of individuals with functional visual loss limits the generalizability of calibrations across the full spectrum of disease severity.

Line 197-8: ‘the relationship between VI and SE, as well as VI and MMD were examined utilizing chi-squared test.' It is not clear how this aligns with the research questions.

Reply: Thank you for the valuable comment. After careful discussion with the authors, we decide to remove this analysis (eTable 2), as it had redundant information to the criterion validity analysis.

Minor:

Abstract: ‘SER' in results was not expanded before.

Reply: Thank you. In method of the Abstract, line 41, we have expanded the ‘SER' as suggested.

Can the subheadings be written in full rather than abbreviations (e.g. VRQoL, RA)?

Reply: Thank you for the important comment. We have corrected the subheadings as suggested.

Line 157: The description under Rasch analysis needs revision. “Andrich rating scale model was used to analyse the C-IVI data in the RA” is not clear.

Reply: Thank you. We have revised the expression to “Rasch analysis was conducted using the Andrich rating scale model with Winsteps software (version 3.72.3, Chicago, Illinois, USA)”.

Line 158: “To assess the psychometric properties of the C-IVI, response category functioning, scale precision, unidimensionality, scale targeting, and differential item functioning” – this sentence is not complete.

Reply: Thank you. The sentence was revised to “To evaluate the psychometric properties of the C-IVI, we examined response category functioning, scale precision, unidimensionality, scale targeting, and differential item functioning”.

Line 166 – “person readability” should be ‘person reliability'

Reply: Thank you for pointing this out. We have corrected the wrong wording.

Line 192: 1 participant with severe VI was combined with moderate VI group for analysis. However, the eTable 2 corresponding line in text (line 208-209) reports severe VI group instead of moderate.

Reply: Thank you. We have removed eTable 2 based on your previous comment.

Line 297: what is ‘Vsl'?

Reply: Thank you, we have corrected the typo ‘VsI' to ‘VI'.

eTable 1: item 18, ‘fafety' should be ‘safety'

Reply: Thank you for the comment, we have corrected the typo.

---

## [Decision Letter · Decision Letter 1]

7 Sep 2025

A psychometric evaluation of the Chinese Impact of Vision Impairment (C-IVI) questionnaire in an adult cohort with high myopia using Rasch Analysis

PONE-D-25-33208R1

Dear Dr. Lan,

We’re pleased to inform you that your manuscript has been judged scientifically suitable for publication and will be formally accepted for publication once it meets all outstanding technical requirements.

Kind regards,

Andrzej Grzybowski

Academic Editor

PLOS ONE

Additional Editor Comments (optional):

Reviewer #1:

Reviewer #2:

Reviewers' comments:

Reviewer's Responses to Questions

**Comments to the Author**

1. If the authors have adequately addressed your comments raised in a previous round of review and you feel that this manuscript is now acceptable for publication, you may indicate that here to bypass the “Comments to the Author” section, enter your conflict of interest statement in the “Confidential to Editor” section, and submit your "Accept" recommendation.

Reviewer #1: All comments have been addressed

Reviewer #2: All comments have been addressed

2. Is the manuscript technically sound, and do the data support the conclusions?

Reviewer #1: Yes

Reviewer #2: Yes

3. Has the statistical analysis been performed appropriately and rigorously? 

Reviewer #1: Yes

Reviewer #2: Yes

4. Have the authors made all data underlying the findings in their manuscript fully available?

Reviewer #1: Yes

Reviewer #2: Yes

5. Is the manuscript presented in an intelligible fashion and written in standard English?

Reviewer #1: Yes

Reviewer #2: Yes

6. Review Comments to the Author

Reviewer #1: Thanks for addressing all my previous comments satisfactorily. The revised manuscript is much improved.

Reviewer #2: (No Response)

7. PLOS authors have the option to publish the peer review history of their article (what does this mean? ). If published, this will include your full peer review and any attached files.

**Do you want your identity to be public for this peer review?** For information about this choice, including consent withdrawal, please see our Privacy Policy .

Reviewer #1: No

Reviewer #2: No
